# Addressing Planetary Health through the Blockchain—Hype or Hope? A Scoping Review

**Rita Issa** [1,2,3,*], **Chloe Wood** [2,4], **Srivatsan Rajagopalan** [2,3], **Roman Chestnov** [5], **Heather Chesters** [6] and **Geordan Shannon** [2,3,4]

1   School of Global Development, University of East Anglia, Norwich NR4 7TJ, UK
2   United Health Futures, 25, Rue Kléberg, 1201 Geneva, Switzerland; srivatsan.rajagopalan.20@ucl.ac.uk (S.R.)
3   Institute for Global Health, University College London, London WC1N 1DP, UK
4   STEMA, 116 High Holborn, London WC1V 6RD, UK
5   Sustainable Development and Internet Studies, Webster University, 1293 Geneva, Switzerland
6   Institute of Child Health Library, Great Ormond Street, London WC1V 6RD, UK
*   Correspondence: r.issa@uea.ac.uk

**Abstract:** Planetary health is an emergent transdisciplinary field, focused on understanding and addressing the interactions of climate change and human health, which offers interventional challenges given its complexity. While various articles have assessed the use of blockchain (web3) technologies in health, little consideration has been given to the potential use of web3 for addressing planetary health. A scoping review to explore the intersection of web3 and planetary health was conducted. Seven databases (Ovid Medline, Global Health, Web of Science, Scopus, Geobase, ACM Digital Library, and IEEE Xplore) were searched for peer-reviewed literature using key terms relating to planetary health and blockchain. Findings were reported narratively. A total of 3245 articles were identified and screened, with 23 articles included in the final review. The health focus of the articles included pandemics and disease outbreaks, the health of vulnerable groups, population health, health financing, research and medicines use, environmental health, and the negative impacts of blockchain mining on human health. All articles included the use of blockchain technology, with others additionally incorporating smart contracts, the Internet of Things, artificial intelligence and machine learning. The application of web3 to planetary health can be broadly categorised across data, financing, identity, medicines and devices, and research. Shared values that emerged include equity, decentralisation, transparency and trust, and managing complexity. Web3 has the potential to facilitate approaches towards planetary health, with the use of tools and applications that are underpinned by shared values. Further research, particularly primary research into blockchain for public goods and planetary health, will allow this hypothesis to be better tested.

**Keywords:** planetary health; global health; blockchain; web3; public goods

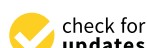

## 1. Introduction

Our understanding of human health and wellbeing, and its drivers, is in constant evolution. Once the sole domain of the individual—our genetics and biochemistry—our approach to population health has iterated from how people act and interact (public health) to wider interconnected communities (global health), human-animal interaction (one health), and into a current paradigm where human health is understood to be interdependent with the well-being of the planet and our ecosystems (planetary health). COVID-19 revealed how humans and the environment are increasingly connected in a fragile "supersystem" that undermines rather than supports planetary health, and characterised planetary health challenges in terms of their complexity, interconnectivity, and chronicity. Such challenges require systems-wide strategies above and beyond current responses.

Planetary Health is defined as "the achievement of the highest attainable standard of health, well-being, and equity worldwide through judicious attention to the human

systems—political, economic, and social—that shape the future of humanity and the Earth's natural systems that define the safe environmental limits within which humanity can flourish" [1]. It is an emergent, action-oriented, transdisciplinary field that focuses on understanding and addressing anthropogenic impacts on Earth's natural systems and their subsequent health, social and environmental sequelae [2]. Aside from a burgeoning academic discourse, planetary health actions span policy, advocacy, and community building [3–5]. Whilst a range of technology-enabled approaches exist for addressing ecological challenges and health(care) as separate fields [6], technology that bridges these domains (i.e., technology for planetary health) remains relatively underdeveloped. One emergent and potentially transformative approach is blockchain technology, which is surrounded by a degree of hype and scepticism in equal measure.

Blockchains are a distributed ledger technology, allowing data to be recorded and distributed securely across a peer network. Adding to the blockchain requires consensus, meaning that data becomes both decentralised and immutable [7]. The term blockchain is associated and sometimes used interchangeably with cryptocurrency (decentralised digital currency housed on the blockchain) and web3 (referring to the new era of the internet where it is possible to not just read data online ('web1'), or read and write ('web2'), but read, write, and own).

There are certain conceptual similarities between planetary health and blockchain: the former describes a complex system which suffers various co-ordination issues, requiring both the centralisation of certain aspects (such as environmental mitigation responses and early warning systems) while benefiting from localised adaptive actions and responses relevant to communities. Blockchain provides tools that can support the management of complexity and the ability to co-ordinate decentralised entities and centralising interventions. As such, there are emergent examples of blockchain utilisation to such social ends, such as in 'Decentralised Science' and 'Regenerative Financing', which respond to challenges across science and ecology. While various review articles have assessed the use of blockchain technologies in health [6,8–10], primarily exploring how web3 technologies can be leveraged for use in health records [11–13], there has been little consideration of wider socio-political and ecological determinants of health, nor of potential interventions. Little attention has been paid to the nexus of blockchain technology and planetary health, nor to the application of web3 solutions for planetary health challenges. As such, this research (i) systematically surveys the academic literature for examples of the web3 and planetary health interface, and (ii) articulates a preliminary framework of shared principles and objectives between web3 and planetary health to identify areas of potential collaboration.

## 2. Materials and Methods

This scoping review examines the literature on blockchain and planetary health in accordance with the framework described by Munn et al. [14]. The protocol was prospectively registered on the Open Science Framework (DOI 10.17605/OSF.IO/KU3WD). Findings were reported following the Preferred Reporting Items for Systematic Reviews and Meta-Analyses extension for Scoping Reviews (PRISMA-ScR) guidelines [15].

### 2.1. Search Method and Information Sources

Searches were conducted in the following electronic databases using date parameters 1 January 2008 and 20 June 2022: Ovid Medline, Global Health, Web of Science, Scopus, Geobase, ACM Digital Library, and IEEE Xplore. The start date reflects Satoshi Nakamoto's paper on Bitcoin and the first successful implementation of blockchain technology in 2008 [16]. Key terms relating to planetary health and blockchain were informed by previous reviews [6,17], including terms relating to global health, one health, environmental health, healthcare financing, the social determinants of health, population health, health equity, ecology and health, and climate change and health; and blockchain, web3, distributed ledger, and cryptocurrency. Supplement Table S1 provides the full search strategy for each respective database search.

*2.2. Screening Process*

Records were imported to Endnote to remove duplicates. After de-duplication, titles and abstracts were independently screened by two researchers according to the selection criteria using the software Rayyan (https://rayyan.ai/reviews; accessed on 20 June 2022). Full texts were then screened using a similar process, with conflicts resolved until consensus was reached among all authors.

Articles were included which met the following criteria: (i) Studies that report on the relationship between blockchain and planetary health, including climate change and health, interconnectivity and health, ecology and health, OneHealth, indigenous health, health equity, population health, and global health; (ii) studies that are global in focus and/or >1 world region; and (iii) studies that include the use of blockchain technology or cover health using frameworks conceptually similar to planetary health, meaning system-level and system-enabling interventions, population-level focus, and/or focus on common goods. Exclusion criteria included: (i) articles published prior to the advent of applied blockchain technology (1 January 2008); (ii) articles mentioning blockchain in passing but not as a key variable of the study; (iii) articles relating to individual health in absence of reference to planetary health/health as a common good; (iv) conference proceedings and studies that do not provide access to the full text; and (v) studies not otherwise meeting the inclusion criteria.

*2.3. Data Extraction and Synthesis*

Data was extracted from all included papers using a pre-piloted data extraction form, iterated by the research team, which included: author, year of publication, type of study/source, methodology used, country/region of focus, health focus, relevance for planetary health, technologies described, key points, limitations, and themes. We used narrative synthesis methodology to analyse and report data. This method of synthesising findings from multiple studies relies primarily on the use of words and text to summarise and explain findings, which is particularly useful where there is heterogeneity of themes, topics, and methods across studies. Blockchain-related terms we were expecting to encounter are outlined in Box 1.

**Box 1.** Blockchain terms.

- Cryptocurrency: a form of currency that only exists digitally and uses a decentralised system to record transactions and manage the issuance of new units, relying on cryptography to prevent counterfeiting and fraudulent transactions [18].
- DApp: 'decentralised apps', offering a similar function to regular apps (such as a phone app/application), but run in a decentralised manner through a peer-to-peer network, such as the blockchain [19].
- Internet of Things/Internet of Medical Things (IoT/IoMT): devices which are connected to the Internet and/or to other connected devices. The 'Internet of Medical Things' is an amalgamation of medical devices and applications that can connect to healthcare information technology systems using networking technologies [20]; for example, a wearable mobile health device.
- Non-fungible tokens (NFTs): a unique digital identifier that cannot be copied, substituted, or subdivided, that is recorded in a blockchain, and that is used to certify authenticity and ownership (as of a specific digital asset and specific rights relating to it) [21].
- Smart contract: a self-executing contract with the terms of the agreement between buyer and seller directly written into lines of code [22].

## 3. Results

From the search, 3245 articles were identified for screening. After de-duplication, 1826 articles were screened by title and abstract by two reviewers. A total of 171 articles were screened in full-text, and 23 articles were ultimately included in the review [Figure 1].

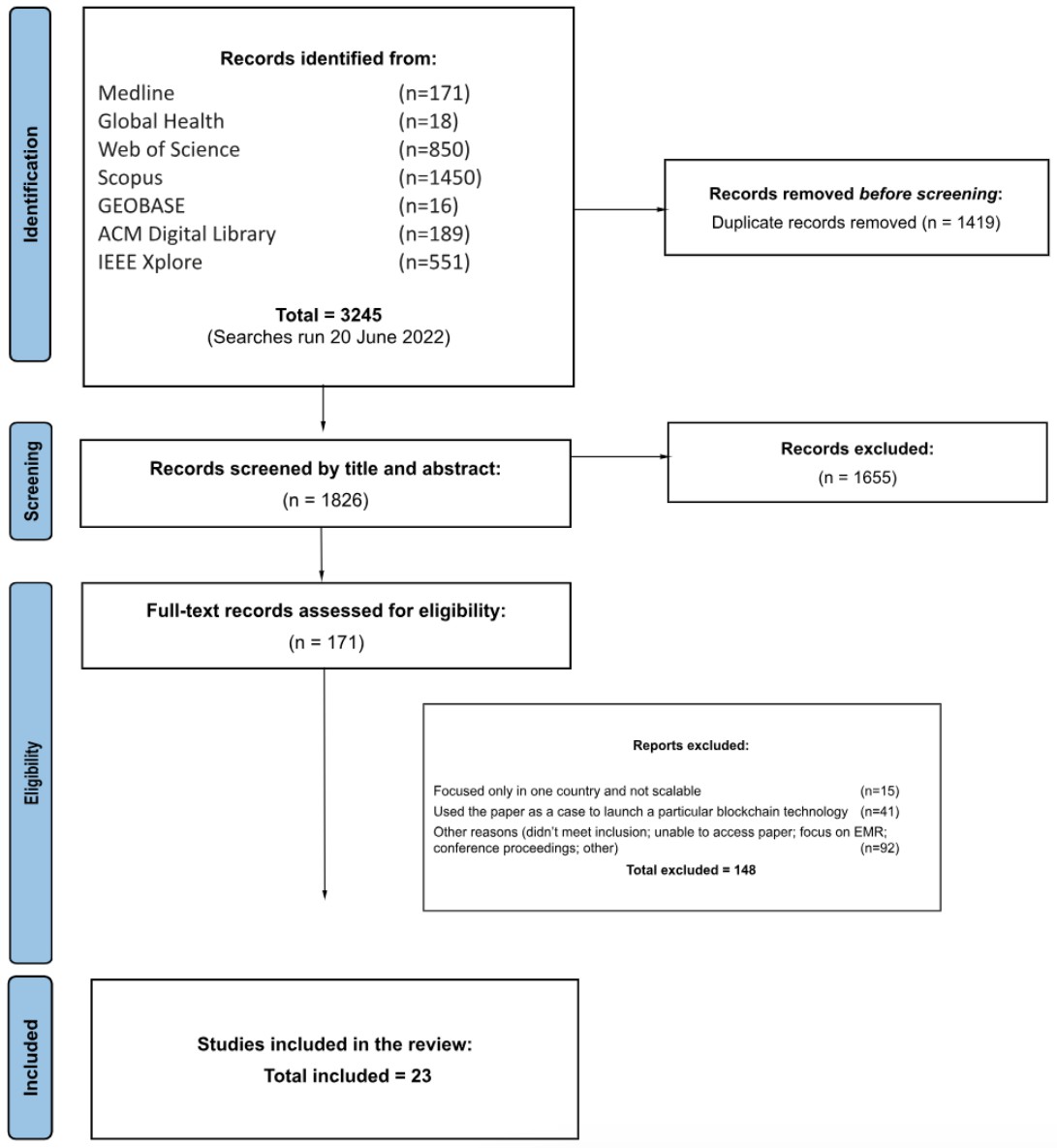

**Figure 1.** Flow Diagram of Search.

### 3.1. Characteristics of Included Publications

All studies were published between 2017 and 2022. Their authors were from institutions based in the USA (n = 11) [23–33], Canada (n = 2) [34,35], China (n = 2) [36,37], India (n = 2) [34,38], Iran (n = 1) [39], Iraq (n = 1) [39], Korea (n = 1) [40], Kyrgyzstan (n = 1) [40], Malaysia (n = 1) [39], the Netherlands (n = 1) [28], Pakistan (n = 1) [40], Portugal (n = 1) [41], Taiwan (n = 1) [42], and Saudi Arabia (n = 1) [43]. The papers were literature or narrative reviews (n = 10) [26,28,29,35,37–40,43,44], conceptual frameworks (n = 10) [23–25,27,30,31,34,36,45] (of which one included expert interviews) [32], a scoping review (n = 1) [41], a development study (n = 1) [42], and a quantitative research study (n = 1) [33]. Summary characteristics are provided in [Table S1 Supplementary Materials].

### 3.2. Health and Technology Characteristics

The health focus of the articles encompassed: pandemics and outbreaks (n = 8), specifically with regards to COVID-19 [39,44], vaccination [42], public health surveillance [27,34,35], and pandemic preparedness [40]; and the use of blockchain to support vulnerable groups health (n = 3), including for homelessness [23], refugees and migrants [41],

and tackling the misuse of opioids [32], alongside specific disease groups (n = 1; HIV [30]). Five articles discussed blockchain use in population health, including to study the social determinants of health and health equity [24], and for population-level health interventions explored across all the WHO health building blocks [26,31,43,45]. Other health applications included financing to support universal health coverage (n = 1) [25]; use in research (n = 2) [28,35]; managing drug development and supply chains (n = 2), including financing orphan drugs [36] and combating the trade in falsified drugs [37]; and application at the environment and health interface (n = 2), with regard to how the environment impacts human health [38] and the negative consequences of cryptocurrency mining on human health [33].

All articles included the use of blockchain technology (BCT), either in isolation (n = 16) [23,27–32,34,35,37–39,41,43–45] or with reference to the use of smart contracts (n = 6) [24–26,36,40,42], the impacts of mining (n = 1) [33], or other technologies, including the Internet of (Medical) Things (IoT/IoMT) (n = 5) [35,38–40,43], artificial intelligence [38], and machine learning [40]. Climate change and/or environment was explicitly mentioned in four papers [33–35,38].

*3.3. Narrative Review of Findings*

A visual summary of the results is shown in Figure 2. Blockchain can be applied to practical planetary health challenges and use cases in a number of ways, across data, financing, identity, medicines and devices, and research. Values-based considerations that emerged in the data include equity, decentralisation, transparency and trust, and managing complexity.

**PLANETARY HEALTH AND BLOCKCHAIN: PRACTICAL APPLICATION AND SHARED VALUES**

**Applications**

**Data**
- Improved interoperability and reduced siloes
- Actionable outcomes, across environmental monitoring, pandemic preparedness & health system responsiveness

**Research**
- Open research: data collection, storage and sharing
- Scientific publishing

**Medicines & Devices**
- Better supply chain efficiency
- Reduction of sub-standard/ falsified drugs and devices

**Financing**
- Improved health insurance efficiency and access
- Improved grant-making and donor funding

**Identity**
- Identity protections reducing barriers to health and social services for marginalised groups

**Values**

Equity

Transparency and trust

Decentralisation

Managing Complexity

**Figure 2.** Key findings and classifications showing practical applications and shared values between planetary health and blockchain.

3.3.1. Practical Applications

Data

For patient interfacing services, open data can support the protection of health information [26,40], the reduction of siloed health services [35,38,41], and the challenges of

data sharing and interoperability issues [35,38]. Patients having access to past records, and interconnected records owned by patients instead of centralised health providers, increases patient control and enables a holistic approach to health [31,35]. Data also supports inputs into health systems, from the individual to the planetary level, particularly by integrating with IOT/IoMT (defined in Box 1). This includes, for example, ambient assisted living for older people [35], remote patient monitoring, environment monitoring for extreme weather events [35,38], and uses during outbreaks [31,34]. Open and transparent data networks can support wider processes for health in LMICs with health system strengthening [25], and around wider factors regarding the social determinants of health [24]. Practically, decentralised, non-siloed data can support telehealth services [26], pandemic preparedness [34], data sharing and storage for national, state, and local health agencies [34], and reducing costs caused by duplication and inefficiencies [41].

Financing

The blockchain broadly can support the financing of population health at multiple levels: individuals interacting with health services, for example through health insurance [30,45]; between different health care stakeholders; for policy makers; and for traditional grants and donor agencies [25]. At the individual, clinic, and health insurance levels, blockchain supports improved cash flow, lower transaction costs, reduced settlement times [43], and efficiency [45]. These qualities, along with the immutability, decentralisation, and transparency of BCT, facilitate trust: between healthcare stakeholders [45], and for individuals who might otherwise usually be excluded from traditional models of financing, such as "homeless youth or other marginalised communities" [30]. Within grant-making and donor agencies, cryptocurrencies can help shift sovereignty to local policymakers, while smart contracts can 'trigger' increased funding when preset goals are achieved, removing the need for intermediaries [25]. This automation of social outcome-based funding adds in new accountability to development financing, including for microfinance and microinsurance, and can support the diversification of funding streams through, for example, impact bonds [25]. BCT also has the potential to revolutionise the financing and reimbursement of intellectual property and drug development by pooling patients or "would-be patients", for example, the financing of orphan drug development through massive group insurance that could be facilitated using BCT due to the lower costs of administration and access to authenticated and secure data [36].

Identity

Lack of proof of identity can limit access to health and other social services. This is particularly challenging for people who are otherwise marginalised, including homeless people, migrants, and refugees [23,41]. BCT can support the development of an immutable digital proof of identity, which alongside health data consolidation, transparency, and global surveillance, allows service providers to direct funding and prevent fraud [25,41].

Medicines and Devices, including Counterfeits and Supply Chains

All stages of the development of medicines and devices for use in healthcare have the potential to be integrated with BCT. Particular focus is given to supply chains, where inefficiencies, theft, and counterfeits can be mitigated through the use of smart contracts [26]; BCT's immutability, which allows records of drugs and service flows, including throughout transportation, to be recorded on-chain to offer visibility and timeliness [40,45]; and the use of IoT applications to ensure the credibility of data input into BCT, which can reduce the number of substandard or falsified drugs [37].

Research

A number of potential uses for BCT are proposed for research: data collection, storage, and sharing for research; onboarding patients; and scientific publishing [30]. Where medical records have previously been centralised and disconnected, the decentralisation

and interoperability provided by BCT could allow health records to be used more readily by researchers [32,45]. For patients, consent for health studies can be more readily given, and connecting with other patients with similar concerns or conditions to pool agency and impact in medical trials can be made easier [35,44]. In terms of publication and translating research into practice, the blockchain can reduce inefficiencies around data sharing and intellectual property issues [28] and prevent false information and predatory journals by promoting the traceability of information such as false infographics or manipulated images [26].

### 3.3.2. Values-Based Considerations

#### Equity

Open data, transparency, and decentralisation can support health equity. The pooling of data and resources could allow for drug development that disrupts a pharmaceuticals paradigm which prioritises profits and thus targets only high-income patients [36]. For health in low- and middle-income countries (LMICs), BCT can support countries and populations to capitalise on their health data (by owning their data) for research and innovation, creating a more equitable global market for health data [25]. This transfer of power and ownership supports equity by removing financial institutions as third-party intermediaries and promoting financial inclusion for the traditionally unbanked [25]; this is particularly useful where there is corruption or instability, and for otherwise marginalised or excluded groups, such as migrants and refugees [41], sex workers [30], and the homeless [23]). BCT can provide tools to support equity beyond proximal health tools, such as by validating and storing important documents, for example, academic degrees [41], and can facilitate the process of sending remittances by reducing intermediaries and transaction times [41].

#### Decentralisation

Shifting the locus of information from centralised bodies, where it is often inaccessible or owned by third parties, and into a decentralised network supports innovation and provides benefits. Centralised stores are vulnerable to attack and limited with regards to security, privacy, and scalability [39,45]. Within specific use cases, decentralised technologies allow peer-to-peer coordination, for example, in energy trading [35,38] and for pooling data or finances for shared aims [36].

#### Transparency and Trust

Blockchain fundamentally ensures a 'trustless' system, meaning transactions can be performed with a basis of trust regardless of any prior relationship. This enables the immutability of data and accountability among strangers [29]. For planetary health considerations, this means the management of variables that may otherwise reduce trust in healthcare services: for example, reducing falsified and substandard drugs [37], validating health records and vaccine cards [13,42], and monitoring health system financing and aid [25].

#### Managing Complexity

BCT can support complexity where different inputs and actors can be coordinated and monitored, and plans executed, in a distributed ledger. This can apply to multi-stakeholder-funded projects [25]; to people on the move [41]; and across different agencies, such as those responsible for housing, food, financial stability, and social support [24]. Where such complexity usually translates into inefficiencies, duplication, and slow coordination and mobilisation, BCT allows better predictions and modelling [38], and more rapid decision-making with faster (sometimes automatically deployed) intervention [44].

#### Cautions

However, such technologies must be deployed with caution and awareness of their limitations. There must be an awareness of the risk of 'crypto-colonial' practices which

mimic past colonial activities in global health and that act in the interests of western capital [26]. The energy consumption of BCT technologies and the impact of mining on human health remain of concern [30,33,41]. Technological and regulatory limitations across health, environment, and blockchain persist, including issues of legal compliance [28], limited technical expertise [30], clinical standards, and data interoperability [30,43], meaning that "if not introduced carefully, blockchain risks overpromising and under-delivering in healthcare" [31].

## 4. Discussion

To our knowledge, this scoping review provides the first comprehensive, systematically obtained overview of planetary health and blockchain. Seven databases were searched to explore evidence of the use or potential use of blockchain in planetary health. The findings can be broadly categorised into practical applications of the blockchain in planetary health and values-based alignments between the two fields.

Planetary health is a broad field encompassing human–environment interaction and its impact on human health, alongside wider socio-political and other drivers of health, climate change, and environmental degradation. As such, there is no one clear path to advancing planetary health. This review highlights a number of practical approaches that might support this wider aim across data, financing, medicines and devices, and research.

BCT is well established and best known for its use in data and financing, the mainstay of blockchain technologies to date. As a distributed ledger that allows data to be recorded and distributed securely across a peer network, the blockchain has already been applied across a number of fields which rely on and benefit from the use of data. For the health sector, it must be noted that this review excluded the vast majority of papers that pertained to electronic health records, as they focused on the individual and/or hospital-level interventions. However, it would be an oversight to not acknowledge the significant weight of literature concerning the use of blockchain in electronic health records, and that managing electronic health records using blockchain could serve some use in advancing planetary health. At the level of public goods and systemic applications and interventions, blockchain application is somewhat sparse, with emerging examples across environmental monitoring, early warning systems, and supply chains [46,47]. An important consideration is how blockchains receive, relay, and act on data. The rise of the 'Internet of Things' or the 'Internet of Medical Things' has been one such tool: devices ranging from electronic watches to weather stations that capture data and relay it into the blockchain, which can be pre-programmed to trigger code or a response through a smart contract. For planetary health, this might mean that automated code could deploy catalytic financing, early warning systems, evacuation warnings, or public health messaging when triggered by certain thresholds in data, such as dangerous temperatures or pollution levels.

The use of blockchain in financing is perhaps its most well-known application to date. The blockchain opens up novel forms of financing, from cryptocurrencies that might be used for specific purposes to NFTs (non-fungible tokens) which denote digital ownership. NFTs have been deployed by a number of charities within and beyond web3 to raise funds; an NFT might be sold as artwork or to publicly assign ownership of an asset, such as trees or forests [48]. Beyond cryptocurrencies and NFTs, blockchain has further potential uses in planetary heath financing. The field has emerged from and aligns closely with the aid sector and philanthropic organisations; while such institutions have significantly advanced global health, they are also beset by bureaucracy, opacity, and neocolonial approaches to financing [49]. The blockchain can facilitate transparency of aid and philanthropic financing, allow for transparent monitoring of impact, reduce unnecessary bureaucracy, and decentralise decision-making power. The growing regenerative financing ('ReFi') community within web3 is closely exploring the idea of financing public goods through the blockchain, where BCT is leveraged to address global sustainability challenges (such as planetary health) and build viable alternatives to the institutions that perpetuate the status quo [50–52].

Open and transparent systems that facilitate the monitoring of goods over time and space have also proven of value in supply chains across a number of sectors, including food, raw materials, and medicines and devices, including for COVID-19 vaccines and in tackling antimicrobial resistance [53]. With regard to the development of drugs and devices, the emergence of the decentralised science ('DeSci') field within web3 has seen organisations developing medicines and devices through an open, collaborative approach beyond traditional, profit-driven incentives [54,55].

### 4.1. Values Alignments between Web3 and Planetary Health

The literature suggests 'aspirational' or values-based alignments between web3 and planetary health, where the web3 technology could support and embed processes that support values that are integral to achieving planetary health. Both population health and climate change are inequitable and unjust. As such, tackling planetary health, whilst not replicating the structures that have caused both health and ecological crises, requires a new approach. One such approach is through decentralisation, moving power away from centralised institutions to impacted people and communities. The blockchain is supporting the development of a new form of organisation, Decentralised Autonomous Organisations (or 'DAOs'), each with their own rules and structures that facilitate non-hierarchical organising by setting predetermined rules into code, thus facilitating trust and collective decision-making.

### 4.2. Limitations of the Current Blockchain Ecosystem

Shortcomings and challenges exist alongside the potentials of the blockchain. Web3, coded by humans and limited by people's worldviews, knowledge, and preconceptions, risks perpetuating or replicating current structures across monetary systems, politics, and other hierarchies and forms of oppression. This is somewhat the case with certain cryptocurrencies, which are beginning to mirror massive wealth inequalities seen within the traditional economic system. Whilst BCT facilitates transparency, the technology is also highly secure and anonymous, potentially supporting tax evasion or corruption. Much BCT use in the health sector is focused on individualised health, while there is still limited mobilisation of web3 for public goods. While BCT is technically open-access, there are barriers to entry, requiring both tech infrastructure (such as a computer) and tech literacy. The technologies are not yet at the stage where they are sufficiently user-friendly for the vast majority of people. Given the barriers to entry, BCT benefits accumulate in already wealthy and privileged health systems and settings [56]; this is reflected in this review, in which half of the surveyed publications were led by authors at North American institutions. Finally, there remain legitimate concerns around the ecological footprint of BCT and cryptocurrency mining, though newer blockchains are utilising different cryptographic approaches to significantly reduce energy expenditure.

### 4.3. Reflections on and Limitations of the Research Landscape

For the purposes of this paper, our working definition of planetary health was expanded beyond articles that explicitly focus on the environment and/or climate change in the context of health to include papers that spoke to conceptual similarities with planetary health: largely systems-level and system-enabling interventions, population-level focus, and/or a focus on common goods. This was necessary due to the very limited literature, just four papers, which addressed the nexus of environment, health, and blockchain. Of these papers, two were similar both in name and content [35,38]. The vast majority of papers were conceptual, review, or synthesis papers, which drew on both grey and academic literature to state their findings. Only two papers drew on primary research (expert interviews and using data), with nine papers referring to case studies of blockchain projects for health. Papers which were country specific, which were excluded from this review, drew far more on primary and secondary data, and case studies. Interestingly, the only article which drew on primary data was also the only paper to frame the blockchain-health-

environment interface critically, with regards to the negative health and environmental externalities of cryptocurrency mining. As such, while a good number of articles were included in the review, few were of particularly high quality. It is important to note that we excluded papers, even those with rigorous literature synthesis, that went on to present an idea for a new blockchain technology or 'DApp'. This was because these papers could not be guaranteed to offer critical and objective reviews of the literature in the context of pitching or promoting a technology.

**5. Conclusions**

The findings of this paper point towards shared values between planetary health and web3–encompassing equity, decentralisation, transparency and trust, andcomplexity, and applications of web3 for planetary health- including for data use, financing, identity verification, medicines and devices, and research. There remains scope for better understanding of how and where web3 may applied to planetary health, both through academic research and testing of use-cases. The public response to blockchain ranges from excitement to scepticism. To date, the most common usage of the blockchain has been in decentralised finance, where despite the promise of otherwise, some cryptocurrencies have replicated economic inequalities and/or 'Ponzi' economic schemes. However, as evidenced in our findings, the scope of blockchain technology is expanding beyond financial markets, such as the emergence of numerous DAOs operating across a range of thematic areas, and the movements for regenerative finance and decentralised science. Planetary health is a challenging field—complex and multifactorial—and addressing planetary health requires shifts across many intersecting systems: human (political, economic, and relational) and planetary. One way to support these shifts is by re-imagining how things are done, by whom, and for what purpose; web3 may offer some of the tools required technically actualise such endeavours.

**Supplementary Materials:** The following supporting information can be downloaded at: https://www.mdpi.com/article/10.3390/challe15010003/s1, Table S1: Summary characteristics of the included articles; Table S2: Full search strategy for the database search.

**Author Contributions:** R.I. conceived the presented idea and developed the research protocol with support from G.S., H.C. supported the development of the search strategy and retrieved the studies from electronic databases. R.I., C.W., S.R. and R.C. participated in the title/abstract screening, full-text screening, and data extraction. R.I. drafted the initial tables, introduction, methods, supplement, results and discussion. Feedback and intellectual input to the manuscript was offered by all authors. All authors have read and agreed to the published version of the manuscript.

**Funding:** The author(s) received no financial support for the research, authorship, and/or publication of this article.

**Data Availability Statement:** All data analysed in this study is included in the publication or Supplement Materials.

**Acknowledgments:** The authors would like to acknowledge Theo Brainin for his comments on the manuscript draft.

**Conflicts of Interest:** R.I., G.S. and S.R. are co-organisers of 'Planet.Health', a voluntary not-for-profit initiative supported by United Health Futures for blockchain and planetary health communities to share learning and co-develop solutions for planetary health.

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
