# Peer review of "Addressing Planetary Health through the Blockchain—Hype or Hope? A Scoping Review"

_challenges, doi:10.3390/challe15010003_

Round 1

Reviewer 1 Report

Comments and Suggestions for Authors

The authors propose a promising research topic for evaluating the perspectives of web3 technologies for addressing planetary health. The review includes a comprehensive literature review covering potential factors that converge to this primary topic.

However, Section 3 relates preponderantly to population health rather than the more encompassing topic of planetary health. This distinction and further classification should be better emphasized in order to highlight the various aspects involved.

Figure 2 presents complementary perspectives: shared values and practical applications. However, the representation should be better described to provide more understanding of the various factors and their correlation. Moreover, there should be more insights regarding how these factors relate to planetary health.

Subsection 1.3.1 should provide some overall context on the practical applications that were analyzed (e.g., how were these classified, and how are they represented in the analyzed research works). Same observation for 1.3.2 - there should be a paragraph to correlate with the overview presented in Figure 2

In terms of paper structure, Box 1 could be removed and the terms included in a subsection focused on conceptual models and methodologies.

Table 1 should be restructured as paragraphs regarding related work (a separate chapter could be created). The main concern is that the content does not effectively provide a comprehensive overview in a tabular format. Key findings and classifications could still be emphasized, more concisely in a table. The results should be included in the Results section instead of the Materials and Methods section, to improve overall readability.

Section 2.3 Data extraction and synthesis is insufficiently described in terms of the materials and methods used (e.g., which are the algorithms, technologies, frameworks, and infrastructures)

Section 2.4 should be removed (these mentions are usually added at the end of the paper)

Subsections with more than 3 levels could be avoided by removing the numbering (e.g., 1.3.1.1, 1.3.1.2).

Comments on the Quality of English Language

Minor editing of English language required

Author Response

Reviewer comments 1

The authors propose a promising research topic for evaluating the perspectives of web3 technologies for addressing planetary health. The review includes a comprehensive literature review covering potential factors that converge to this primary topic.

However, Section 3 relates preponderantly to population health rather than the more encompassing topic of planetary health. This distinction and further classification should be better emphasized in order to highlight the various aspects involved.

In 2.2 Line 143 we state: “Studies that report on the relationship between blockchain and planetary health, to include: climate change and health, interconnectivity and health, ecology and health, OneHealth, indigenous health, health equity, population health, and global health”.

We then later address this in section 4.3 Line 434 – stating “For the purposes of this paper, our working definition of Planetary Health was expanded - beyond articles that explicitly focus on environment and/or climate change in the context of health - to include papers that spoke to conceptual similarities within planetary health: largely systems-level and system-enabling interventions, and/or population level focus  and/or a focus on common goods. This was necessary due to the very limited literature - just 4 papers - which addressed the nexus of environment, health and blockchain (i.e. ‘planetary health’).”

Figure 2 presents complementary perspectives: shared values and practical applications. However, the representation should be better described to provide more understanding of the various factors and their correlation. Moreover, there should be more insights regarding how these factors relate to planetary health.

Per the reviewers point below that “Key findings and classifications could still be emphasized, more concisely in a table. The results should be included in the Results section instead of the Materials and Methods section, to improve overall readability” – Figure 2 attempts to summarise key findings and classifications. As such, the figure content and title has been updated to present this information.

Subsection 1.3.1 should provide some overall context on the practical applications that were analyzed (e.g., how were these classified, and how are they represented in the analyzed research works). Same observation for 1.3.2 - there should be a paragraph to correlate with the overview presented in Figure 2

Under section 2.3 I have added:

This method of synthesising findings from multiple studies relies primarily on the use of words and text to summarise and explain findings, particularly useful where there is heterogeneity of themes, topics and methods across studies.”

In terms of paper structure, Box 1 could be removed and the terms included in a subsection focused on conceptual models and methodologies.

Thank you. We have moved Box 1 into section 2.3 of the methods.

Table 1 should be restructured as paragraphs regarding related work (a separate chapter could be created). The main concern is that the content does not effectively provide a comprehensive overview in a tabular format.

“Table 1” follows the usual reporting format for scoping reviews, whereby summary characteristics of all studies are included in tabular form for ease of review. This usually does not include ‘summary findings’ but we have added this to ensure a ‘quick read’ overview in tabular format.

Key findings and classifications could still be emphasized, more concisely in a table. The results should be included in the Results section instead of the Materials and Methods section, to improve overall readability.

I am not sure what this refers to as Table 1 (summary characteristics) and Figure 2 (Key findings) are both included in the ‘Results’ section 3, not in ‘materials and methods’.

Section 2.3 Data extraction and synthesis is insufficiently described in terms of the materials and methods used (e.g., which are the algorithms, technologies, frameworks, and infrastructures)

We used narrative synthesis methodology to analyse and report data. This method of synthesising findings from multiple studies relies primarily on the use of words and text to summarise and explain findings, particularly useful where there is heterogeneity of themes, topics and methods across studies.

That said, we have added Box 1 definitions – stating “Blockchain-related terms we were expecting to encounter are outlined in Box 1”.

The subsequent narrative findings (which outline the algorithms, technologies, frameworks from the data) were not predetermined, and are described in section 3.2 of the results, line 277 onwards.

Section 2.4 should be removed (these mentions are usually added at the end of the paper)

This has been removed.

Subsections with more than 3 levels could be avoided by removing the numbering (e.g., 1.3.1.1, 1.3.1.2).

Thank you – we have removed the numbering with more than 3 levels and updated this to correctly refer to the numerical ordering of the document.

Minor editing of English language required

Thank you, we have proof-read the paper and updated the language and grammar where required. If further editing is required do let us know.

Reviewer 2 Report

Comments and Suggestions for Authors

The manuscript proposes an interesting literature review on the nexus of blockchain technology and planetary health.

The manuscript is well written and organized, and the proposed issues can be of interest to the journal’s target audience.

However, before considering it for publication, some improvements are needed.

In the introduction, research motivations should be stressed in order to better highlight the potential benefits of the proposed research. Moreover, at the end of this section, the article route could be briefly illustrated.

In section 2, the role of table 1 and the criteria used to determine the search strategy are unclear. Hence, this point must be clarified by elaborating more on the search criteria, the definition of the keywords used in the search strings of the selected databases, etc. This is the major criticality of the manuscript and to solve it you can refer to the following studies, where a detailed description of the definition of search criteria is well explained: https://doi.org/10.1080/09537287.2015.1129464; https://doi.org/10.3390/buildings10060098. The reference to the above studies (and if you need to other similar successful literature review articles) can certainly allow you to define a scientifically sound research approach.

In section 3, the numbering of subsections should be revised. Moreover, the quality of Figure 1 could be augmented: e.g. the use of a sans serif font could help its readability.

I appreciated the discussion section very much and I think that following the above suggestions can certainly lead to a high-quality literature review article.

Author Response

Reviewer comments 2

The manuscript proposes an interesting literature review on the nexus of blockchain technology and planetary health.

The manuscript is well written and organized, and the proposed issues can be of interest to the journal’s target audience.

However, before considering it for publication, some improvements are needed.

In the introduction, research motivations should be stressed in order to better highlight the potential benefits of the proposed research. Moreover, at the end of this section, the article route could be briefly illustrated.

Thank you for this feedback. We have added in a section in the introduction (line 68 onwards) explaining the rationale and motivations for undertaking this research:

“There are certain conceptual similarities between Planetary Health and Blockchain: the former describes a complex system which suffers various coordination issues, re-quiring both centralisation of certain aspects (such as environmental mitigation responses and early warning systems) while benefiting from localised adaptive actions and re-sponses relevant to communities. Blockchain provides tools that can support management of complexity and the ability to coordinate decentralised entities and centralising inter-ventions. As such, there are emergent examples of Blockchain utilisation to such social ends, such as in ‘Decentralised Science’ and ‘Regenerative Financing’ which respond to challenges across science and ecology. While various review articles have assessed the use of blockchain technologies in health [6,13–15]; primarily exploring how web3 technologies can be leveraged for use in health records [16–18], little consideration to wider socio-political and ecological determinants of health, nor potential interventions. There is scarce synthesis on the nexus of blockchain technology and planetary health, nor the application of web3 solutions for planetary health challenges.”

In section 2, the role of table 1 and the criteria used to determine the search strategy are unclear. Hence, this point must be clarified by elaborating more on the search criteria, the definition of the keywords used in the search strings of the selected databases, etc. This is the major criticality of the manuscript and to solve it you can refer to the following studies, where a detailed description of the definition of search criteria is well explained: https://doi.org/10.1080/09537287.2015.1129464; https://doi.org/10.3390/buildings10060098. The reference to the above studies (and if you need to other similar successful literature review articles) can certainly allow you to define a scientifically sound research approach.

Supplement table 1 and the search strategy follows the protocol described by Munn et al.

The text has been updated to state:

Key terms related to planetary health and blockchain were informed by previous reviews [6,21], including terms relating to global health, one health, environmental health, healthcare financing, the social determinants of health, population health, health equity, ecology and health, and climate change and health; and blockchain, web3, distributed ledger and cryptocurrency. Supplement Table 1 provides the full search strategy for each respective database search.

In section 3, the numbering of subsections should be revised. Moreover, the quality of Figure 1 could be augmented: e.g. the use of a sans serif font could help its readability.

Thank you – the numbering has been corrected and the figure updated.

I appreciated the discussion section very much and I think that following the above suggestions can certainly lead to a high-quality literature review article.

Thank you!

Round 2

Reviewer 1 Report

Comments and Suggestions for Authors

The authors have satisfactorily improved the manuscript and it is now better organized. Hence, it can be considered for publication. It needs some minor revisions to correct some formatting errors, which should be addressed in the final edited version. Some of the provided answers referenced sections that do not exist (e.g. 3.2, which should be numbered accordingly). The figure labels should be positioned below the figures (not above). 

Reviewer 2 Report

Comments and Suggestions for Authors

The Authors have satisfactorily improved the manuscript. Hence, it can be considered for publication